# Integrative Analysis of Proteomics and Metabolism Reveals the Potential Roles of Arachidonic Acid Metabolism in Hypoxia Response in Mouse Spleen

**DOI:** 10.3390/molecules27228102

**Published:** 2022-11-21

**Authors:** Yujing Guo, Sheng Yong, Yuzhen Xu, Ying Hu, Jidong Li, Qifu Long, Xiaojun Wang, Cunlin Gu, Zengqiang Miao

**Affiliations:** Medical College of Qinghai University, Xining 810016, China

**Keywords:** spleen, arachidonic acid, proteomics, metabolism, immune response

## Abstract

High altitude hypoxia stress is the key cause of high-altitude pulmonary edema and spleen contraction. The molecular mechanism of immune response of various tissue systems to hypoxia stress remains lacking. In this study, we applied proteomics combined with metabolomics to explore the key molecular profilings involved in high altitude hypoxia response in the spleen of mice. The results showed that 166 proteins were significantly up-regulated, and only 39 proteins were down-regulated. Bioinformatics analysis showed that mineral absorption, neuroactive ligand–receptor interaction, arachidonic acid metabolism, IL-17 signaling pathway and NOD-like preceptor signaling pathway were significantly enriched in the list of 166 upregulated differentially expressed proteins (DEPs). Among these metabolic pathways, the former three pathways were co-identified in KEGG terms from LC-MS/MS based metabolic analysis. We further found that both arachidonate 15-lipoxygenase and hematopoietic prostaglandin D synthase were upregulated by around 30% and 80% for their protein levels and mRNA levels, respectively. Most downstream metabolites were upregulated accordingly, such as prostaglandin A2 and D2. This study provides important evidence that arachidonic acid metabolism potentially promotes spleen hypoxia response through a combined analysis of proteomics and metabolism, which could bring new insights for the spleen targeted rational design upon arachidonic acid metabolism of new therapies.

## 1. Introduction

High-altitude hypoxia exerts a great impact on the physical and mental health of plateau residents. It is known that when the human body rises to a plateau area above 3000 m, the partial pressure of oxygen in the atmosphere will drop sharply with the increase in altitude, which results in hypoxia of tissue cells. As part of these changes, oxygen demand often exceeds supply and can result in overall oxygen deficits [1,2,3,4,5]. This condition indeed seriously affects the body’s metabolism and normal physiological functions, including acute altitude sicknesses such as high altitude pulmonary edema (HAPE) and high altitude cerebral edema (HACE) [1,6,7]. This stressful condition could also exacerbate risk of cystemia and various chronic cardiovascular diseases [2]. The changes in the body under high altitude hypoxia exposure and the molecular mechanisms of various tissue systems in response to hypoxia stress have become the subject of high-altitude medical research in recent years.

The immune system is an important defense system of the body. The immune system plays an irreplaceable role in maintaining the homeostasis of tissues. Accumulative studies reported that hypoxia regulates the proliferation, development and effector function of immune cells through transcriptional regulation driven by hypoxia inducible factors (HIFs) [3,4]. It has been reported that hypoxia could increase the immune cell infiltration and inflammatory response in various organs of mice, and hence leads to the occurrence of acute altitude sickness [5,8]. Therefore, the relative impact of high-altitude hypoxic environment on the body immune system urgently requires widespread attention.

Arachidonic acid (AA) is an important polyunsaturated fatty acid, which represents a wide range of immune-physiological effects in animals. As an extremely important structural lipid, it widely exists in mammalian organs, muscles and blood. For example, in tissues, it is an essential fatty acid in animals. Arachidonic acid is the precursor of various biologically active substances [9]. There are mainly three types of enzymes involved in the metabolism of arachidonic acid. The oxidation reactions that catalyze the metabolism of arachidonic acid mainly include: epoxidation, allyl oxidation and Ω hydroxylation [10]. AA is catalyzed by lipoxygenase (LPO) to generate hydroxyeicosatetraenoic acid (HETEs), leukotrienes (LT) and lipoxins (LXs) [11], and LPO is a dioxygenase [12]. The LPO metabolites of arachidonic acid (HETEs and their precursors HPETEs and LTs) have inhibitory effects on immune cells and immune responses. In addition, the synthesis of arachidonic acid as a substrate under the catalysis of enzymes has biological activity. In this regard, given the arachidonic acid substrate is reduced or replaced, this will directly lead to the change in some hypoxia induced regulators, resulting in a series of inflammatory and immune cell responses [13]. However, the metabolic pathways of arachidonic acid are complex, and it is difficult to identify the key factors that affect the immune response, and the physiological function of arachidonic acid in different stressful conditions remain unclear [11].

In recent years, multi-omics combined analysis technology has been gradually applied to the study of key factor analysis of complex biological pathways. Various effective protein factors, biomarkers and anticancer factors were successfully identified through combined analysis of transcriptome, proteome and metabolome. For example, based on proteomic and transcriptomic integrative analysis, Wang and colleagues reported that total m6A levels of mRNAs were reduced during hypoxia, possibly mediated by induction of the m6A eraser ALKBH5, and they suggested that crosstalk between m6A and HIF1 pathways is important for cells to be deficient in hypoxia [14]. In addition, integrative analysis of proteomics and metaboliomics provides the potential to identify changes in protein and metabolites levels following hypoxic exposure and may provide important insights into the mechanisms involved in hypoxic preconditioning [15,16]. The reaction of oxygen is crucial. Through combined proteomic and metabolomic analysis, some scholars have reported the effect of KDAC inhibition (KDACi) on the metabolic profile, and pointed out that the up-regulation of glycolysis, TCA cycle, oxidative phosphorylation and fatty acid synthesis has become a common metabolic response of KDACi [17]. Through combined transcriptome and metabolome analysis, it was found that amino acids not only act as potential osmotic agents, but also contribute to energy generation in response to hypoxia in the clam hypoxia response [18].

In this study, we focused on identifying which key molecular factors are involved in immunological responses in the hypoxia response. With this purpose, we applied a combined proteomic and metabolomic analysis to identify co-altered metabolic and biological pathways. The key differential proteins and/or metabolites in the metabolic pathway of arachidonic acid will be obtained to describe the molecular mechanism that affects the immune response of kidney organs, which is beneficial to expand our knowledge on main physiological functions of arachidonic acid in different organs, and bring new insights for the spleen targeted rational design upon arachidonic acid metabolism of new therapies.

## 2. Results 

In this study, we first performed a TMT-based proteomics analysis, and results show that there are 72,824 peptides detected and these peptides were mapped to 7727 proteins (Appendix A). The peptides numbers declined gradually following the protein coverage from 0.1~1.0 (Appendix A). The cumulative fractions increased dramatically when coefficient of variation were between 0.1 to 0.4, and appear to be plateau when CV was more than 0.4 for either hypoxia group or normalxia group (Appendix A). According to subcellular localization analysis, the 32.97% proteins were annotated to be related to nuclear proteins accounting for total 7727 proteins (Appendix A). The length values for the most peptides were ranged from 8 to 16 (Appendix A). The protein mass ranges majority at from 30~60 kDa (Appendix A). The most mass spectrum shows desirable precursor ion tolerance (Appendix A). The ratio of unique peptides increased gradually when the peptide numbers were less than 28 (Appendix A). This evidence suggests that the proteomics are successfully performed in this study.

According to GO and KEGG analysis, we found that most proteins are related to oxidation–reduction process and protein phosphorylation. Around 227 and 197 proteins are related to nucleus and integral component of membrane. Most proteins are related to protein binding and ATP binding based on molecular function classification (Appendix A). In addition, the proteins annotated were involved in many primary metabolic processes or their relevant derivate metabolic pathways, such as carbon hydrate metabolism, immune system, environmental adaptation, transport catabolism and signal transduction (Appendix A). In addition, we found that there are 166 upregulated differentially expressed proteins (DEPs) induced by hypoxia treatment in the spleen of mice, which is four times more than the numbers of downregulated DEPs (Figure 1A; Appendix A). These DEPs contain some known hypoxia markers that are upregulated in the hypoxia treatment mice, including Hmox1, Trf, Thbs1, Itgb2 and Cd47 (Figure 1A; Appendix A). All DEPs were visualized based on the heatmap indicated in Figure 1B.

Among the 166 upregulated DEPs, we further performed the GO and KEGG analysis to found the altered biological pathways and metabolic pathways. Results show that proteolysis, carbohydrate derivative catabolic process, hydrolase activity and GTPase activity were significantly enriched in the list of these proteins (Figure 2A). In addition, we found that mineral absorption, neuroactive ligand–receptor interaction, arachidonic acid metabolism, IL-17 signaling pathway and NOD-like preceptor signaling pathway were significantly enriched in the list of 166 upregulated DEPs (Figure 2B).

Therefore, we are interested to carry out an integrative analysis of proteomic and non-targeted metabolism to found if there are some overlapped metabolic pathways that co-occurred in both analyses. From the metabolomic analysis, we identified 4285 downregulated differentially abundant metabolites (DAMs) and 4026 upregulated DAMs (Figure 3A), while there are 74 and 76 downregulated DAMs and upregulated DAMs with annotation by metabolic library (Appendix A). Next, we attempted to compare the overlapped metabolic pathways that significantly enriched in the list of DEPs and DAMs. Result show that there are 57 overlapped metabolic pathways that co-occurred in the list of DEPs and DAMs (Figure 3B; Appendix A; Appendix A). The top four KEGG terms that were both significantly enriched in the list of DEPs and DAMs include arachidonic acid metabolism, amobiasis, neuroactive ligand–receptor interaction and mineral absorption (Figure 3C).

Furthermore, we compared the expression levels of proteins and metabolites involved in arachidonate acid metabolism, and found that arachidonate 15-lipoxygenase (Alox15), hematopoietic prostaglandin D synthase (Hpgds), prostaglandin G/H synthase (Ptgs2) and phospholipase D (Pld3) were significantly upregulated regarding their protein abundance from proteomics analysis (Figure 4A). Except for the arachidonic acid (peroxide free) (M303T39) and 1-stearoyl-2-oleoyl-sn-glycerol 3-phosphocholine (M855T142_1), 20-hydroxyarachidonic acid (M303T70), 15-deoxy-delta-12,14-PGJ2 (M315T42), 8-iso-prostaglandin A2 (M333T42) and prostaglandin D2 (M335T120) were all up-regulated by hypoxia treatments (*p* < 0.05) (Figure 4B). Then the qPCR experiments were conducted to determine the mRNA expression levels. Results show that the expression levels of *Alox15* and *Hpgds* were upregulated by one and four times resulted by hypoxia treatments, respectively (Figure 4C,D). The expression levels of *Ptgs2* gene were slightly enhanced but not significantly altered due to hypoxia effects (Appendix A). Consistently, the protein expression of Hpgds, Alox15 and Ptgs2 were increased by at least two times based on Western blot experiments (Appendix A).

As summarized by Figure 5, membrane phospholipids were converted to arachidonate acid catalyzed by Pld3, and the two forms of latter metabolite (arachidonate acid), i.e., arachidonate acid (peroxide free, M303T39) and 20-hydroxyarachidonic acid (M303T70), as detected in current study were not altered or significantly enriched by hypoxia treatments, and they can be converted to both prostaglandin G2 and 5-HPETE by PGH synthase and Alox15, respectively. Hypoxia stimulates expression of some genes related to hypoxia induced factors (HIFs) such as Hmox1 (heme oxygenase 1, ENSMUSP00000005548.6), which could participate in the prostaglandin G2 biosynthesis (Appendix A). The prostaglandin G2 was than sequentially converted to prostaglandin H2 and other derivatives such as PGFol, PGD2, PGA, PGJ2 and TXA. PDG2 (M335T120), PGA (M333T42) and PGJ2 (M315T42) as detected by metabolomic analysis show significantly enriched, and these metabolites are hypothetically helpful to promote immune response to hypoxia stress.

## 3. Discussion

The T lymphocyte immune response and pathological mechanism related to polycythemia caused by high altitude hypoxia remains unclear. The spleen is the most important temporary peripheral immune organ for T lymphocytes, and the size and physiological function of the spleen can affect the number and immune function of T lymphocytes to a certain extent [19]. High-altitude hypoxic conditions often lead to spleen contraction, which affects the normal function of the immune system [20]. In the present study, comparative proteomics and LC-MS-based metabolomics were applied to investigate proteins and metabolite changes in mice spleen in response to short-term hypoxia at the whole-organism level. We found the some overlapped metabolic pathways, in particular, arachidonic acid metabolism, between proteins and metabolites, which suggest the main metabolic pathways during the acute hypoxia stress. These findings provide the hypothetical models at the proteins and metabolic levels for further investigation of mice spleen in response to acute hypoxia.

### 3.1. Arachidonic Acids Involved in T Cell Immune Response

Arachidonic acids are a class of unsaturated fatty acids with 20 carbon atoms. Platelets as well as most tissues in the body have arachidonic acid metabolism, which plays an important role in many different physiological and pathological states. In this study, we found that arachidonic acid could be involved in the hypoxic response of the spleen through multi-omics analysis, indicating that arachidonic acid may be related to the physiological process of T lymphocytes. For example, arachidonic acid-regulated calcium signaling in T cells, which could promote synovial inflammation response [21]. Interestingly, we found the metabolite related to calcium signaling was declined, probably due to the fact that some calcium channel component is AA-regulated calcium-selective (ARC) channel, some calcium channels might be inhibited by leukotriene C_4_ through phosphorylation [22]. It was reported to reduce T-cell responses through inducing mouse dendritic cells maturation [23]. Consistently, we observed that a key metabolite, 1-stearoyl-2-arachidonoyl-sn-glycerol (M628T137) involved in T cell receptor signaling pathway (C00165) was significantly decreased due to hypoxia effects (Appendix A; Appendix A).

### 3.2. Hypoxia Induced Accumulation of Arachidonic Acids Deteriotives

When living organisms experiences acute hypoxia sickness or a bacterial infection, a suite of brain-mediated responses occur, including fever, anorexia and sleepiness. Systemic administration of lipopolysaccharide could increase body temperature, hence induces the production of pro-inflammatory prostaglandins (PGs), including PGE2 and PGD2 [24]. PGD2 is a major inflammatory mediator produced by mast cells and Th2 cells. In current study, we found that the contents of PGD2 were increased by 1.2 times due to hypoxia stimulation (M335T120, *p* < 0.05), suggesting that inflammatory prostaglandins increased by hypoxia through lipopolysaccharide (Appendix A). Hematopoietic prostaglandin D-synthase (Hpgds) is responsible for the formation of PGD2 from PGH2. We found that the both mRNA expression and protein expression of Hpgds were upregulated by 80% and 30%, respectively (Figure 4; Figure 5; Appendix A). The protein expression of Hpdgs was further validated by Western blot (Appendix A). Therefore, Hpgds is probably responsible for the production of PGD2 in the peripheral tissues and in immune and inflammatory cells of mice spleen. Notably, we performed the multi-omics combined analysis using mice spleen exacts. There might be other protein changes if using directly spleen lymphocytes.

### 3.3. Energy Metabolism Promotes Immune Response by Hypoxia

Hypoxia-inducible factor (HIF) is a central transcription factor that enables adaptive response to hypoxic stress in normal and pathological conditions by activating a large number of genes responsible for oxygen delivery, angiogenesis, cell proliferation, cell differentiation and metabolism [25]. We found that a key enzyme (heme oxygenase 1, ENSMUSP00000005548.6) in HIFs signaling pathway was increased by 1.3 times resulted by hypoxia (Appendix A), suggesting hypoxia treatment effectiveness. In addition, HIFs plays a central role in the adaptive regulation of energy metabolism, by triggering a switch from mitochondrial oxidative phosphorylation to anaerobic glycolysis in hypoxic conditions. As found by KEGG analysis on the list of DAMs, we found many energy metabolic pathways were enriched, such as the Pentose phosphate pathway (map00030), Carbon metabolism (map01200) and Glycolysis/Gluconeogenesis (map00010) (Appendix A). Hypoxia also induces oxidoreductase systems, for example, the conversion of L-Glutamate (M148T395) into L-Glutamine (M191T369) involved in Glyoxylate and dicarboxylate metabolism by hypoxia (Appendix A), suggesting that HIFs could reduce oxygen consumption in T-cell, activating autophagy of mitochondria concomitantly with reduction in reactive oxygen species production [26].

## 4. Materials and Methods

### 4.1. Experimental Animals and Experiment Grouping

SPF-grade C57BL/6 healthy male mice aged 6–8 weeks were used to avoid the effects of periodic fluctuations of physiological index and the hormone levels due to estrogen in female animals. All methods were performed in accordance with the relevant guidelines and regulations. All methods are reported in accordance with ARRIVE guidelines. The mice samples were grown with standard food and water way of feeding and environmental control. Before the hypoxia treatments (HST), the mice are randomly divided into 2 groups with 5 mice for each group. The control group is the normoxia group (PSC) and HST group were raised in the LACMD (400 m above sea level) and in the Maduo County People’s Hospital of the Guoluo Tibetan autonomous prefecture (MCPH), Qinghai Province Laboratory animal room (4200 m above sea level), respectively. The two groups of mice were fed routinely in cages in a controlled environment with a temperature of 18–22 °C and a humidity of 45–55%. After hypoxia treated 30 days, the mouse spleens were collected, and one part was fixed and stored in 10% neutral formalin solution and the other part was frozen and stored in liquid nitrogen for later use.

### 4.2. Proteomics Analysis

#### 4.2.1. Protein Extraction

Each sample was ground individually in liquid nitrogen and lysed with PASP lysis buffer (100 mM NH_4_HCO_3_, 8 M Urea, pH 8), as documented previously, followed by 5 min of ultrasonication on ice [27]. The lysate was centrifuged at 12,000× *g* for 15 min at 4 °C and the protein concentration was measured by BCA protein assays (Beyotime, Shanghai, China). The denatured proteins in denaturing buffer were reduced with 10 mM DTT for 1 h at 56 °C, and subsequently alkylated with sufficient IAM for 1 h at room temperature in the dark. Trypsin (Promega, Madison, WI, USA) and 100 mM TEAB buffer were added, sample was mixed and digested at 37 °C for 4 h. Formic acid was mixed with digested sample, adjusted pH under 3, and centrifuged at 12,000× *g* for 5 min at room temperature. The supernatant was slowly loaded to the C18 desalting column, washed with washing buffer (0.1% formic acid, 3% acetonitrile) 3 times, then eluted by some elution buffer (0.1% formic acid, 70% acetonitrile).

To improve the accuracy of the quantitative determination of the identified peptides, we performed tandem mass tag (TMT) labeling LC-MS/MS, coupled to a parallel proteomics monitoring (PRM) based targeted proteomics approach. An amount of 100 μL of 0.1 M TEAB buffer was added to reconstitute, and 41 μL of acetonitrile-dissolved TMT labeling reagent was added, then the sample was mixed with shaking for 2 h at room temperature. Then, the reaction was stopped by adding 8% ammonia. Mobile phase A (2% acetonitrile, adjusted pH to 10.0 using ammonium hydroxide) and B (98% acetonitrile) were used to develop a gradient elution.

For transition library construction, shotgun proteomics analyses were performed using an EASY-nLCTM 1200 UHPLC system (Thermo Fisher, Waltham, MA, USA) coupled with a Q Exactive HF-X mass spectrometer (Thermo Fisher) operating in the data-dependent acquisition (DDA) mode. A 1 μg sample was injected into a home-made C18 Nano-Trap column (4.5 cm × 75 μm, 3 μm). The full scan range was from *m*/*z* 350 to 1500 with resolution of 60,000 (at *m*/*z* 200), the automatic gain control (AGC) target value was 3 × 106 and the maximum ion injection time was 20 ms. The top 40 precursors of the highest abundant in the full scan were selected and fragmented by higher energy collisional dissociation (HCD) and analyzed in MS/MS.

The Maxquant search engine (v.1.6.6.0) was used to analyze the resultant data, with MS/MS spectra being searched against the Mus_musculus UniProt database (https://www.uniprot.org/proteomes; accessed on 15 February 2019). The mass tolerance for precursor ions was set as 20 ppm in First search and 5 ppm in Main search, respectively, and 0.02 Da was set for fragment ions. FDR was adjusted to <1% and minimum score for peptides was set >40. The mass spectrometry proteomics data have been deposited to the ProteomeXchange Consortium [28] via the PRIDE partner repository with dataset identifier PXD029980.

#### 4.2.2. The Functional Analysis of Protein

Gene Ontology (GO) was conducted using the interproscan program against the non-redundant protein InterPro database (https://www.ebi.ac.uk/interpro; accessed on 23 July 2019) [29], and KEGG (Kyoto Encyclopedia of Genes and Genomes) were used to analyze the protein family and pathway. DEPs were used for volcanic map analysis, cluster heat map analysis and enrichment analysis of GO and KEGG [30].

#### 4.2.3. Metabolic Dete rminations

Added to ~25 mg samples was 100 μL lysis buffer to homogenize, and the mixture was then vortexed for 60 s, 400 ul methanol-acetonitrile solution (1:1, *v*/*v*) was added, followed by low temperature ultrasonic for 30 min. The mixture was centrifuged at 2000× *g* and 4 °C for 10 min, thereafter filtered with 0.43 μm organic phase medium (GE Healthcare, 6789-0404). The samples were separated by an Agilent 1290 Infinity LC ultra-high performance liquid chromatography (UHPLC) HILIC column (Yokohama, Japan). Electrospray ionization (ESI) (Thermo Fisher, Bremen, Germany) was used for detection in both positive and negative ion modes. The samples were separated by UHPLC and analyzed by mass spectrometry using an Agilent 6550 mass spectrometer (Waltham, MA, USA). ESI source conditions are as follows: Gas Tem: 250 °C, Drying gas: 16 L/min, Nebulizer: 20 psig, Sheath gas Tem: 400 °C, sheath Gas Flow: 12 L/min, Vcap: 3000 V, Nozzle voltage: 0 V. Fragment: 175 V, Mass Range: 50–1200, Acquisition rate: 4 Hz, cycle time: 250 ms.

The metabolomic profiling was performed with a metabolon software (Durham, NC, USA) as described previously [31]. The sample components were identified by comparing the retention time and mass spectra with those for reference metabolites. For the sample metabolic compounds identification, we used the mass spectra with the entries of the mass spectra libraries NIST02 and the Golm metabolome database (http://csbdb.mpimp-golm.mpg.de; accessed on 13 September 2019).

#### 4.2.4. qPCR Experiments

The extracted total RNA as mentioned above from same spleen tissue was used for Real-time PCR (qPCR). The reverse transcription of RNA was performed with reverse transcription kit (TaKaRa), and qPCR experiments were conducted using SYBR Green real-time PCR kit (TaKaRa). β-actin1 gene (Forward: 5′-GCC CTA GAC TTC GAG C-3′; reverse: 5′-CTT TAC GGA TGT CAA CGT-3′) was used as an internal reference. The 2^-ΔΔCT^ method was used to determine the fold change in gene expression in the HST relative to PSC. The primer sequences are shown in Appendix A. The RT-qPCR reaction system is 20µL in total, and the reaction conditions are: 95 °C for 30 s; 95 °C for 5 s, 60 °C for 60 s; and 40 cycles as described earlier [32]. Three biological replicates were conducted and three technical replicates for each biological replicate were conducted.

#### 4.2.5. Western Blotting Validation

Total protein was extracted from spleens of mice in each group using animal total protein extraction kit (Sangon Biotech, Shanghai, China, C510003). ~50 μg protein was loaded on 12% SDS-PAGE and transferred to nitrocellulose membrane (PALL, P/N66485, Waltham, USA). The membrane was sealed with 5% skim milk for 2 h and incubated with primary antibodies at 4 °C overnight. After washing the membrane with TBST buffer, the membrane was incubated with the secondary antibody at room temperature for 90 min. ECL developer solution was formulated (Thermo, VG299080) and imaging was performed using a gel image analysis imaging system (Fusion Solo, Vilber Lourmat, France). The antibodies for Hpgds (22522-1-AP) and Ptgs2 (66351-1-Ig) were purchased from Proteintech (Shanghai, China), while the antibody of Alox15 (Ab244205) was purchased from Abcam (Shanghai, China).

## 5. Conclusions

High altitude hypoxia stress causes high altitude pulmonary edema and spleen contraction. The molecular mechanism of immune response of spleen to hypoxia stress remains lacking. In this study, we applied a combined analysis of proteomics and metabolomics to identify the key molecular profilings and/or pathways involved in high altitude hypoxia response in the spleen of mice. We found that 166 and 39 proteins were upregulated and downregulated, respectively. Based on the combined analysis, we found mineral absorption, neuroactive ligand–receptor interaction, arachidonic acid metabolism were significantly enriched in the list of both DEPs and DAMs. The arachidonate 15-lipoxygenase and hematopoietic prostaglandin D synthase were upregulated by around 30% and 80% for their protein levels and mRNA levels, respectively. This study provides an insight into novel biological function of arachidonic acid metabolism in spleen hypoxia response.

## Figures and Tables

**Figure 1 molecules-27-08102-f001:**
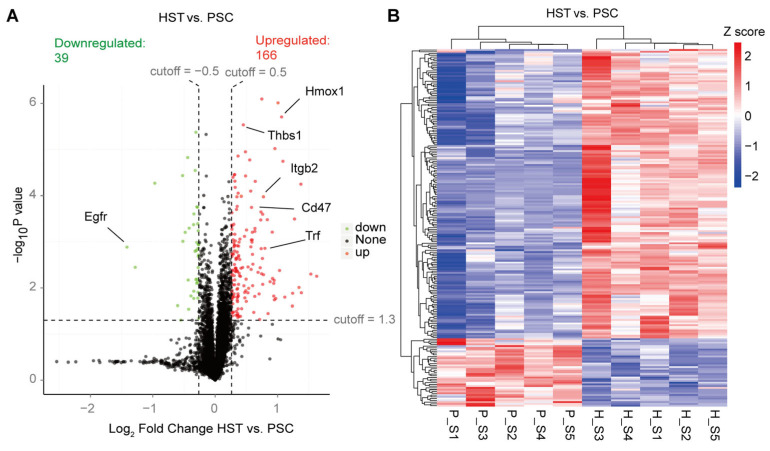
TMT-based proteomic analysis on mice spleen subjected either hypoxia (HST) or normaxia (PSC) conditions. (**A**) Volcano analysis representing the significantly differentially expressed proteins (DEPs). (**B**) Heatmap representing the visualized relative Z scores for each DEP induced by hypoxia treatments. The Z scores were indicated in different colors of cells. The detailed information of protein, e.g., Egfr, was referred to Appendix A.

**Figure 2 molecules-27-08102-f002:**
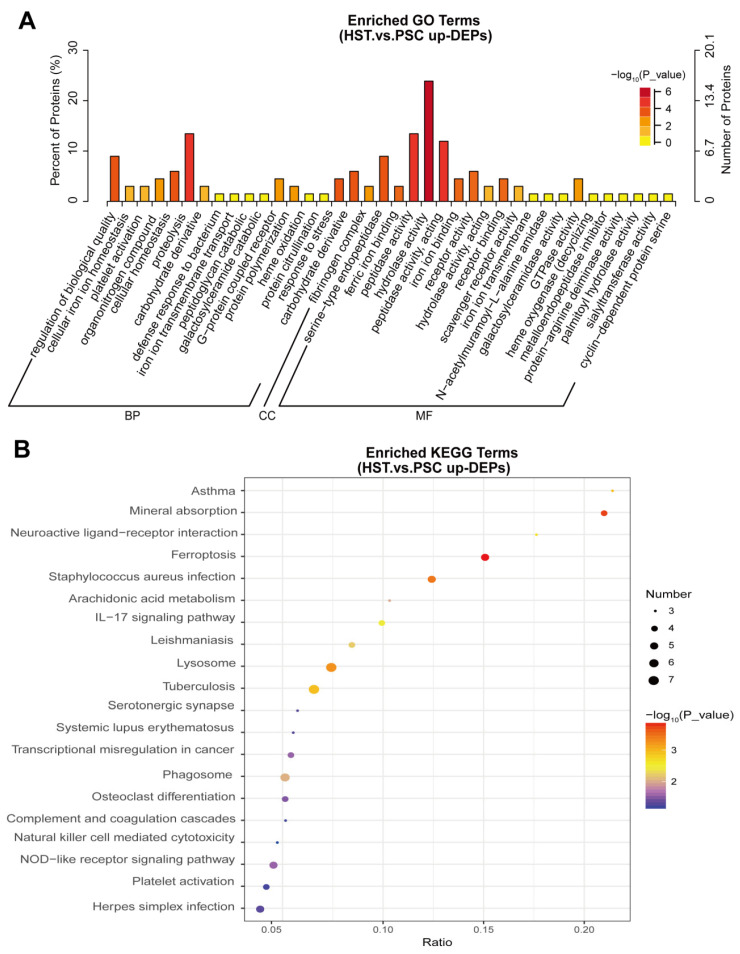
GO and KEGG based on upregulated differentially expressed proteins (DEPs) annotated based on proteomics analysis using spleen samples of mice subjected either by normaxia or hypoxia treatments. (**A**), GO analysis. (**B**), KEGG analysis. BP, CC and MF represent biological pathway, cellular components and molecular function, respectively.

**Figure 3 molecules-27-08102-f003:**
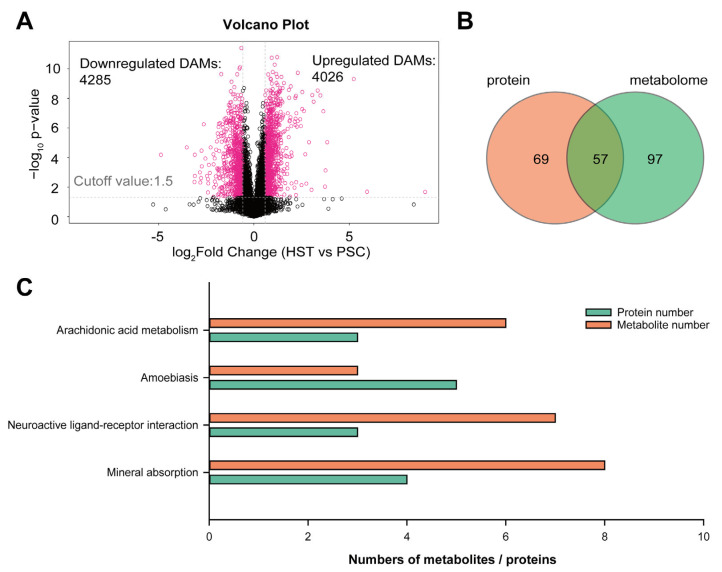
Integrative analysis on proteomics and metabolism using the spleen of mice subjected to either hypoxia and normaxia conditions. (**A**), Volcano plot representing the differentially abundant metabolites induced by hypoxia treatments. (**B**), Venn diagram representing the overlapped KEGG terms on DEPs and DAMs. (**C**), Overlapped top 4 KEGG terms on the list of DEPs and DAMs. DAMs, HST and PSC represent differentially abundant metabolites, hypoxia and normaxia, respectively. In panel (**A**), significant DAMs were represented in pink color.

**Figure 4 molecules-27-08102-f004:**
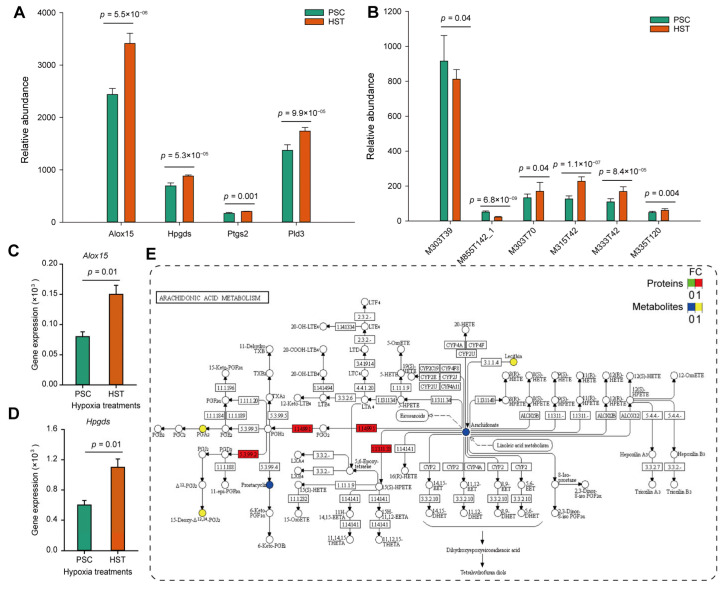
Working model representing the potential mechanism that arachidonate acid metabolism involved in hypoxia response. (**A**), Abundance of four proteins related to arachidonate acid metabolism based on proteomics dataset. (**B**), Relative abundance of metabolites related to arachidonate acid metabolism based on metabolism dataset. (**C**, **D**), Relative expression of three genes related to arachidonate acid metabolism based on qPCR experiments. (**E**), Working model representing the biological roles of arachidonate acid metabolism involved in hypoxia response in mouse spleen. Different colors stand for the relative values of HST vs PSC for either protein or metabolite abundances. Alox15: arachidonate 15-lipoxygenase; Hpgds: hematopoietic prostaglandin D synthase; Ptgs2: Prostaglandin G/H synthase; Pld3: phospholipase D. arachidonic acid (peroxide free) (M303T39); 1-stearoyl-2-oleoyl-sn-glycerol 3-phosphocholine (M855T142_1); 20-hydroxyarachidonic acid (M303T70); 15-deoxy-delta-12,14-PGJ2 (M315T42); 8-iso-prostaglandin A2 (M333T42); prostaglandin D2 (M335T120). For panels **A–E**, each bar of data represents the mean (±SE) of 3 different biological replicates. In panel (**E**), FC represents fold change.

**Figure 5 molecules-27-08102-f005:**
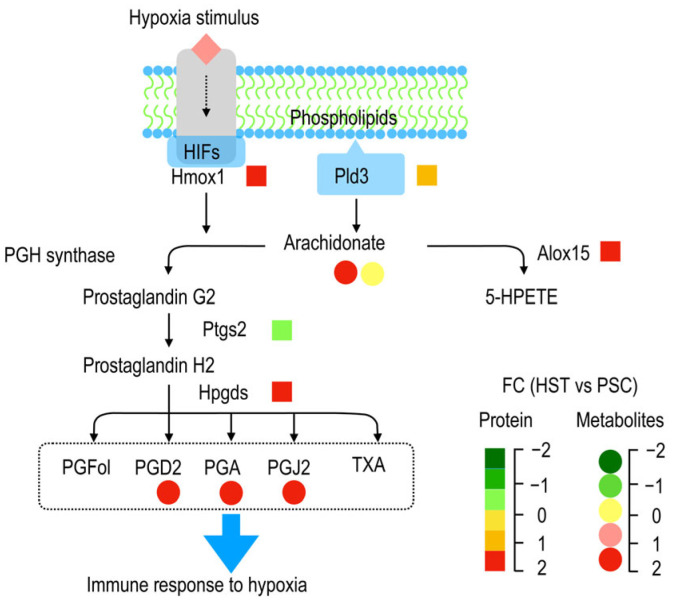
Schematic representation of the changes in metabolites and proteins involved in arachidonate acid metabolism during hypoxia response in spleen of mice. Names of PGFol, PGD2, PGA and PGJ2 were referred to Appendix A. FC represents fold change.

## Data Availability

All data are available in the manuscript or in the Appendix A.

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
