# Peer review of "Integrative Analysis of Proteomics and Metabolism Reveals the Potential Roles of Arachidonic Acid Metabolism in Hypoxia Response in Mouse Spleen"

_molecules, 2022, doi:10.3390/molecules27228102_

Round 1

Reviewer 1 Report

The manuscript by Guo et al. aimed to identify some key molecular factors of immune responses in spleens isolated from mice after the hypoxia treatment. The authors combined proteomic and metabolomic analysis. Proteomic analysis revealed 166 significantly upregulated proteins and only 39 downregulated proteins. GO and KEGG enrichment analyses revealed that the most enriched terms were related to mineral absorption and arachidonic acid metabolism, which seems to play an important role in spleen hypoxia response. Although the obtained results are interesting and are supported by volcano plots or heatmaps, I have several critical remarks:

1) Reference numbers should be in square brackets. Wang et al. (line 73) - reference number is missing. This study contains 27 references only. More references should be cited.

2) Which method was used for the determination of protein concentration in spleen samples (Lowry, BCA..)? This should be described in the methodology section.

3) Why the authors did not try to isolate directly spleen lymphocytes?

4) Why this study included only male mice?

5) The authors should add links to web pages of all stated databases in the text.

6) Although the authors provided supplementary figures, supplementary tables are missing.

7) Limitations of the study are not described and should be added.

8) Caption to Figure 1D: differentially expressed proteins (DAMs) – should be DEPs

9) Figure 3A: Valcano plot – should be Volcano plot

10) line 237: protomics analysis – should be proteomic analysis

Author Response

The manuscript by Guo et al. aimed to identify some key molecular factors of immune responses in spleens isolated from mice after the hypoxia treatment. The authors combined proteomic and metabolomic analysis. Proteomic analysis revealed 166 significantly upregulated proteins and only 39 downregulated proteins. GO and KEGG enrichment analyses revealed that the most enriched terms were related to mineral absorption and arachidonic acid metabolism, which seems to play an important role in spleen hypoxia response. Although the obtained results are interesting and are supported by volcano plots or heatmaps, I have several critical remarks:

1) Reference numbers should be in square brackets. Wang et al. (line 73) - reference number is missing. This study contains 27 references only. More references should be cited.

Reply: Thanks for your comments. The format of reference was corrected to square brackets. In addition, we added three more references in the revised manuscript (refs: Cui et al., 2015 [15], Gao et al., 2016 [16] and Vizcaíno et al., 2014[20]).

2) Which method was used for the determination of protein concentration in spleen samples (Lowry, BCA..)? This should be described in the methodology section.

Reply: Thanks for your comments. We determined the protein concentration using BCA method. This information was added in method section (lines 118-119) in the revised manuscript.

3) Why the authors did not try to isolate directly spleen lymphocytes?

Reply: We agree that directly using spleen lymphocytes might give different protein changes. Basically, there are some realistic limitations due to local harsh experimental condition. The plateau is the on-site environment, 400 kilometers away from Xining. But we are very interested and attempt to perform this experiment in the future study.

4) Why this study included only male mice?

Reply:Thanks for your inquiry. The use of male samples is mainly to avoid the effects of periodic fluctuations of physiological index and the hormone levels caused by estrogen in female animals. The statement was added in the method section (lines 99-101) in revised manuscript.

5) The authors should add links to web pages of all stated databases in the text.

Reply: Revision done.

6) Although the authors provided supplementary figures, supplementary tables are missing.

Reply: Thanks for your gentle reminder. We have now uploaded the supplementary tables.

7) Limitations of the study are not described and should be added.

Reply: Thanks for your comments. We have to say that due to the limitation of some realistic harsh condition, we did not successfully transport the samples used for spleen lymphocytes isolation. There might be some differences in DEPs list by using spleen lymphocytes. We have now added the limitations statement in discussion section in lines 340-342.

8) Caption to Figure 1D: differentially expressed proteins (DAMs) – should be DEPs

Reply: Revision done.

9) Figure 3A: Valcano plot – should be Volcano plot

Reply: Revision done.

Reviewer 2 Report

Previously, the information presented in this paper would have been published in several individual publications. My point is that an avalanche of detailed study data is presented that makes it difficult for the reader (at least for me) to digest it.

very complex presentation of of huge amount of data.

clarification of major an dminor results may be helpful to the reader

Author Response

Reviewer 2:

Previously, the information presented in this paper would have been published in several individual publications. My point is that an avalanche of detailed study data is presented that makes it difficult for the reader (at least for me) to digest it.

very complex presentation of of huge amount of data.

Sorry for the confusion.

 clarification of major an dminor results may be helpful to the reader

The major results:

The minor results:

Reply:

Thanks for your concern about data presentation. We have now made a brief summary including major results and minor results for your review as followings:

Major results:

  1. In this study, we applied a combined analysis of proteomics and metabolomics to identify the key molecular profillings and/ or pathways involved in high altitude hypoxia response in the spleen of mice.
  2. We found that 166 and 39 proteins were upregulated and downregulated, respectively.

Minor results:

  1. Based on the combined analysis, we found mineral absorption, neuroactive ligand-receptor interaction, arachidonic acid metabolism were significantly enriched in the list of both DEPs and DAMs.
  2. The arachidonate 15-lipoxygenase and hematopoietic prostaglandin D synthase were upregulated by around 30% and 80% for their protein levels and mRNA levels, respectively. The results were further validated by western blot.

We sincerely hope that the information above would be helpful for your positive evaluation on our manuscript.

Reviewer 3 Report

In general, you have introduced the topic well with some minor problems. You have mentioned that it was previously reported that hypoxia could increase immune cells infiltration and inflammation response in various organs of rats and hence leads to the occurrence of acute altitude sicknessNevertheless, in your study you used mice. I suggest to include articles regarding mice rather than rats as it might be confusing for the readers.
There are minor typo errors such as:
line 54: "Arachidonic acidis the precursor...." you need a space between acid and is.
line 57: "The oxidation reactions that catalyze the metabolism of arachidonic acidmainly include:....." you need a space between acid and mainly.
In the introduction you focus on describing the importance of the multi-omic technology including transcriptomics, proteomics and metabolomics. Nevertheless, in your study you did not perform transcriptomics. It will be more beneficial for your study to focus in studies where proteomics and metabolomics are used. 

Methods:
Methods sufficiently described for reproducibility with minor issue. Particularly, you do not mention the analysis of the proteomic data; Which software you used to analyse them? For instance, did you use MaxQuant software or any other? What are the parameters of your search? Which database did you use?  Additionally, did you submit your data in PRIDE (Proteomics Identifications Database). You may consider to submit your data in PRIDE. 
Results/Discussion
"In this study, we aimed to elucidate the potential molecular mechanism of hypoxia effects on spleen of mice." In my opinion, it will be better to avoid using the word "molecular" as you do not perform any further validation of your data. You have performed a well-designed experiment in order to investigate the proteome and metabolome of mice spleen, but you do not have any evidence for any molecular mechanism. 
"These DEPs were visualized based on the 199 heatmap indicated in Figure 1E, suggesting that hypoxia treatment promotes most protein expression in spleen of mice." In this sentence you claim that hypoxia treatment promotes protein expression but you do not provide evidence to support this claim. The higher expression of those proteins could be due to protein degradation. In the next sentence you even mentioned that proteolysis is one of the top biological pathways which alter.  Remember protein expression levels depend on the balance on between their synthesis and degradation rates.
In this study, the proteomics were successful. They identified 7727 individual proteins. Personally, I will prefer to move Figure 1A-C to the supplementary materials as they prove the quality of their proteomic but they do not add any value to the scientific question which this study is trying to answer. In figure 1D, it will be more appropriate if you label the Cutoffs used in the volcano blot I am assuming the cutoffs are: -log10 P value 1.3 and log2 fold of change -0.5 and 0.5 but it is not clear.  Finally, it will be better if you show known hypoxia markers that are upregulated in the hypoxia treatment mice (HST). One of the main weakness of this study is the absence of evidence supporting that the HST mice group is under hypoxia. 
Figure 2 has lack of consistency. In the KEGG enriched you have correctly used -log10P_Value but not in the GO terms. Why? You should include the -log10P_Value for GO terms too. 
Figure 3 A please include your cutoff and you may consider to use more harsh cutoffs. Spelling mistake Volcano.
in line 241-242 you mentioned 'Then the qPCR experiments were conducted to validate the protein expression levels'. This statement is wrong. qPCR can be used to measure mRNA levels. You need to re-write this part and make it more clear. 
I could find several spelling mistakes. For example. Figure 4 and Figure 5 titles start with lower case instead of upper case.
Line 237 proteomics instead of protomics.

Overall, this study has a clear biological question which is How mouse spleen changes upon hypoxia response?”. The in vivo experiment was nicely designed and described. The use of 5 biological replicants is a strong point for this study. Additionally, both proteomic and metabolomic were successfully performed. Nevertheless, the analysis is incomplete as it is not mentioned how the proteomic analysis was performed and the data was not submitted in PRIDE. They need to provide the list of those 7727 proteins. Additionally, this study is lacking any further validations except of a qPCR which is an incorrect method of measuring the protein levels. If the authors want to support their data, they should consider to perform some Western blots to prove the upregulation or downregulation of some of their hits. Additionally, the untargeted metabolomic are not as reliable as the targeted metabolomic. They may further validate some of their hits. The authors mention the use of metabolic library (Supplementary table). I did not manage to find this table.
Lastly, the authors should reconsider their title. The study is lacking a strong evidence to support that arachidonic acid metabolism is essential in hypoxia response. In my opinion, according to the data, it seems to be clear that potentially the arachidonic acid metabolism changes during hypoxia but I do not see any evidence regarding its role and importance. This study could be a brilliant resource paper for scientists of this particular areaHowever, this paper needs several improvements mentioned above. 

Author Response

In general, you have introduced the topic well with some minor problems. You have mentioned that it was previously reported that hypoxia could increase immune cells infiltration and inflammation response in various organs of rats and hence leads to the occurrence of acute altitude sicknessNevertheless, in your study you used mice. I suggest to include articles regarding mice rather than rats as it might be confusing for the readers.

Reply: Thanks. We have now changed the rats to mice in main text, and also corrected the word in supplementary files in the revised manuscript to avoid the misleading.

There are minor typo errors such as:
line 54: "Arachidonic acidis the precursor...." you need a space between acid and is.

Reply:Revision done.

line 57: "The oxidation reactions that catalyze the metabolism of arachidonic acidmainly include:....." you need a space between acid and mainly.

Reply: Revision done.

In the introduction you focus on describing the importance of the multi-omic technology including transcriptomics, proteomics and metabolomics. Nevertheless, in your study you did not perform transcriptomics. It will be more beneficial for your study to focus in studies where proteomics and metabolomics are used. 

Reply: Thanks for your constructive comments. We have now added two new references reporting the proteomics and metaboliomics in reference list [15] and [16].

Methods:
Methods sufficiently described for reproducibility with minor issue. Particularly, you do not mention the analysis of the proteomic data; Which software you used to analyse them? For instance, did you use MaxQuant software or any other? What are the parameters of your search? Which database did you use?  Additionally, did you submit your data in PRIDE (Proteomics Identifications Database). You may consider to submit your data in PRIDE. 

Reply: Thanks for your concern on the proteomic analysis process. The information of software and relevant parameters were added in the method section in lines 145-151. In addition, the proteomics data used in this study was uploaded to PRIDE database as recommended.

Results/Discussion
"In this study, we aimed to elucidate the potential molecular mechanism of hypoxia effects on spleen of mice." In my opinion, it will be better to avoid using the word "molecular" as you do not perform any further validation of your data. You have performed a well-designed experiment in order to investigate the proteome and metabolome of mice spleen, but you do not have any evidence for any molecular mechanism. 

Reply: The sentence “we aimed to elucidate the potential mechanism of hypoxia effects on spleen of mice” was removed in the revised manuscript.

"These DEPs were visualized based on the 199 heatmap indicated in Figure 1E, suggesting that hypoxia treatment promotes most protein expression in spleen of mice." In this sentence you claim that hypoxia treatment promotes protein expression but you do not provide evidence to support this claim. The higher expression of those proteins could be due to protein degradation. In the next sentence you even mentioned that proteolysis is one of the top biological pathways which alter.  Remember protein expression levels depend on the balance on between their synthesis and degradation rates.

Reply: Thanks for your constructive comments. We have now removed the clarification “hypoxia treatment promotes most protein expression in spleen of mice” from the revised manuscript to avoid misleading the message.

In this study, the proteomics were successful. They identified 7727 individual proteins. Personally, I will prefer to move Figure 1A-C to the supplementary materials as they prove the quality of their proteomic but they do not add any value to the scientific question which this study is trying to answer.

Reply: Thanks for your constructive comments. The panels A-C in Figure 1 were moved to supplementary figure S2 as recommended.

In figure 1D, it will be more appropriate if you label the Cutoffs used in the volcano blot I am assuming the cutoffs are: -log10 P value 1.3 and log2 fold of change -0.5 and 0.5 but it is not clear. 

Reply: The cutoffs values were added in the volcano plot in Figure 1.

Finally, it will be better if you show known hypoxia markers that are upregulated in the hypoxia treatment mice (HST). One of the main weakness of this study is the absence of evidence supporting that the HST mice group is under hypoxia. 

Reply: We appreciate it for your comments. We have now re-organized the Figure 1, and the known hypoxia markers were added in Figure 1 in volcano plot as advised.

Figure 2 has lack of consistency. In the KEGG enriched you have correctly used -log10P_Value but not in the GO terms. Why? You should include the -log10P_Value for GO terms too. 

Reply: The -log10P_Values were added for GO terms as suggested.

Figure 3 A please include your cutoff and you may consider to use more harsh cutoffs. Spelling mistake Volcano.

Reply: The cutoff values were added in Figure 3A. The spelling mistake “Valcano” was corrected to “Volcano” as well.

in line 241-242 you mentioned 'Then the qPCR experiments were conducted to validate the prote in expression levels'. This statement is wrong. qPCR can be used to measure mRNA levels. You need to re-write this part and make it more clear.

Reply:  The statement was corrected to “Then the qPCR experiments were conducted to determine the mRNA expression levels” in lines 262-263. 

I could find several spelling mistakes. For example. Figure 4 and Figure 5 titles start with lower case instead of upper case.

Reply: Thanks for your comments. The spelling mistakes were corrected in

Figure 4 and 5, we also revised the other parts in the manuscript.

Line 237 proteomics instead of protomics.

Reply: Revision done.

Overall, this study has a clear biological question which is “How mouse spleen changes upon hypoxia response?”. The in vivo experiment was nicely designed and described. The use of 5 biological replicants is a strong point for this study. Additionally, both proteomic and metabolomic were successfully performed. Nevertheless, the analysis is incomplete as it is not mentioned how the proteomic analysis was performed and the data was not submitted in PRIDE.

Reply: Thanks for your positive evaluation on our manuscript. We took your comments into account in the revised version of the manuscript. The information of software and relevant parameters were added in the method section in lines 145-151. In addition, the proteomics data used in this study was uploaded to PRIDE database as recommended.

They need to provide the list of those 7727 proteins. Additionally, this study is lacking any further validations except of a qPCR which is an incorrect method of measuring the protein levels. If the authors want to support their data, they should consider to perform some Western blots to prove the upregulation or downregulation of some of their hits.

Reply: The list of 7,727 proteins was added in Table S2. In addition, as suggested, we have now added the results of western blots experiments and the results reveal that the expression levels of Alox15, Pgds2 and Hpgds involved in arachidonic acid metabolism were enhanced by at least two times, which supports the proteomics data. These results were added in the revised manuscript in lines 266-268.

Additionally, the untargeted metabolomic are not as reliable as the targeted metabolomic. They may further validate some of their hits. The authors mention the use of metabolic library (Supplementary table). I did not manage to find this table.

Reply: We really appreciate it for your constructive comments. This is a very interesting point. We attempted to conduct this experiment, but due to prevalent COVID-19, we encounter many difficulties, 1) the campus was totally blocked; 2) there is limited samples; 3) courier companies closed. The targeted metabolomics analysis would be definitely useful to be performed in the future study. In response to your second inquiry, the supplementary table was uploaded in the submission system.

Lastly, the authors should reconsider their title. The study is lacking a strong evidence to support that arachidonic acid metabolism is essential in hypoxia response. In my opinion, according to the data, it seems to be clear that potentially the arachidonic acid metabolism changes during hypoxia but I do not see any evidence regarding its role and importance. This study could be a brilliant resource paper for scientists of this particular area. However, this paper needs several improvements mentioned above. 

Reply: Thanks for your positive evaluation. The title was changed to “Integrative Analysis of Proteomics and Metabolism Reveals the Potential Roles of Arachidonic Acid Metabolism in Hypoxia Response in Mouse Spleen”.

Round 2

Reviewer 3 Report

Authors took into account the previous comments. The manuscript is improved. I completely understand the problems that pandemic caused to several laboratories worldwide, therefore under this circumstances, I believe manuscript is ready for publication. I have a minor comment regarding figure S6. The western blot analysis indicated the up-regulation of  two proteins related to arachidonate acid metabolism in hypoxia-treated spleen of mice. This figure could  be moved to figure 3 as it is a strong supporting evidence regarding their hypothesis. It will be nicer if these results appear in the manuscript rather than the supplementary material.